# Obesity and Diabetes in Mexico: An Approach to the Intestinal Microbiota

**DOI:** 10.3390/nu17233661

**Published:** 2025-11-23

**Authors:** Ruth Michelle Acosta-Meneses, Esther Ramírez-Moreno, Laura Berenice Olvera-Rosales, Alejandra Cardelle-Cobas, Zuli Guadalupe Calderón-Ramos, Gabriela Mariana Rodríguez-Serrano, Carlos Manuel Franco-Abuín, Alberto Cepeda-Saéz, Luis Guillermo González-Olivares, Alicia del Carmen Mondragón-Portocarrero

**Affiliations:** 1Instituto de Ciencias Básicas e Ingeniería, Área Académica de Química, Universidad Autónoma del Estado de Hidalgo, Carr. Pachuca-Tulancingo km. 4.5, Pachuca 42184, Hidalgo, Mexico; 2Instituto de Ciencias de la Salud, Área Académica de Nutrición, Universidad Autónoma del Estado de Hidalgo, Circuito Ex Hacienda, La Concepción S/N, Carretera Pachuca Actopan, San Agustín Tlaxiaca 42060, Hidalgo, Mexico; esther_ramirez@uaeh.edu.mx (E.R.-M.);; 3Laboratorio de Higiene, Inspección y Control de Alimentos, Departamento de Química Analítica, Nutrición y Bromatología, Campus Terra, Universidade da Santiago de Compostela, 27002 Lugo, Spain; 4División de Ciencias Biológicas y de la Salud, Universidad Autónoma Metropolitana-Iztapalapa, Mexico City AP 55-355, Mexico

**Keywords:** obesity in Mexico, diabetes in Mexico, intestinal microbiota, probiotics, metabolic health

## Abstract

Obesity and diabetes have reached alarming prevalence rates globally, with Mexico being one of the most affected countries. This review explores the epidemiology of these metabolic disorders and analyzes their prevalence and risk factors, as well as the crucial role of the intestinal microbiota in their development. Obesity and diabetes in Mexico have been linked to lifestyle factors, genetic predispositions, and alterations in the gut microbial composition. The intestinal microbiota plays a significant role in metabolic regulation, and its dysbiosis has been associated with insulin resistance, chronic inflammation, and increased fat accumulation. Studies in Mexico have highlighted specific microbial patterns in individuals with obesity and diabetes, suggesting a unique interplay between diet, microbiota composition, and metabolic health. Strategies to restore microbial balance, such as dietary modifications and probiotic interventions, have shown promising results in improving metabolic parameters and reducing disease progression. However, challenges remain in understanding the long-term effects of microbiota-targeted therapies and their individual variability. This review underscores the need for further research to develop personalized interventions aimed at modulating the gut microbiota for obesity and diabetes management. Future directions should focus on integrative approaches combining nutrition, prebiotics, and microbiome-based therapeutics to combat the increasing burden of metabolic diseases in Mexico.

## 1. Introduction

Diabetes and obesity remain significant public health concerns in Mexico [1]. In recent decades, both conditions have reached epidemic levels, impacting a large portion of the population [2]. While this issue is not exclusive to Mexico, the country records one of the highest obesity rates worldwide and a notable prevalence of type 2 diabetes (T2D) [3]. In the past decade, especially during and after the COVID-19 pandemic, Mexico has experienced a marked increase in cases of obesity and diabetes [4]. Data from the 2023 National Health and Nutrition Survey (ENSANUT) reveal that 19.3% of adults aged 20–39 years and 19.9% of those aged 40–59 years are overweight or obese, indicating a steady upward trend. The prevalence of diabetes has also increased, affecting 10.3% of the adult population, with many cases remaining undiagnosed [5]. This public health crisis is fueled by factors such as unhealthy dietary choices, decreased physical activity, heightened sedentary behavior, and genetic factors [6,7]. The link between obesity and diabetes is further intensified by modern lifestyles and urbanization, straining Mexico’s healthcare system [8].

In recent years, research has increasingly focused on the relationships between these diseases and the gut microbiota, providing new insights into their underlying mechanisms and potential therapeutic strategies [9,10,11]. The gut microbiota has been recognized as a crucial factor in the link between diabetes and obesity, with studies demonstrating significant differences in microbial composition between obese individuals and nonobese individuals with obesity, as well as between those with and without T2D [12,13,14].

In individuals with obesity, reduced microbial diversity and an increased Firmicutes-to-Bacteroidetes ratio have been observed, which promote increased calorie absorption and fat accumulation [15]. In T2D, microbial dysbiosis can impair insulin sensitivity and contribute to systemic inflammation, both of which are central to disease progression [16,17]. Additionally, metabolites produced by the microbiota, such as short-chain fatty acids, play crucial roles in regulating blood glucose levels [18].

On the basis of these findings, modulating the gut microbiota through dietary interventions, probiotics, and prebiotics represents a promising strategy for managing obesity and diabetes [19]. This review aims to provide an overview of the impact of obesity and diabetes on Mexican society, explore their relationships with the gut microbiota, and examine relevant studies conducted within the Mexican population. Additionally, potential intervention strategies involving probiotics will be proposed, providing new insights into the treatment and prevention of these conditions.

## 2. Prevalence of Obesity and Diabetes and Risk Factors

### 2.1. Obesity in Mexico

Obesity is a chronic, multifactorial, noncommunicable disease associated with dietary habits and sedentary lifestyles [20]. The World Health Organization (WHO) defines obesity as an abnormal or excessive accumulation of body fat that poses health risks [21]. In adults, classification is based on body mass index (BMI), whereas indirect indicators, such as waist circumference, have been established to evaluate abdominal fat accumulation. The official Mexican Standard 043 sets specific cutoff points for the Mexican population (Table 1) for these measures, which are fully consistent with those established by the WHO for adults. Obesity was divided into three groups (I, II, III), and the same BMI ranges were used. The difference lies in their application: the WHO establishes a universal reference for clinical and epidemiological studies, whereas NOM 043 contextualizes this classification within dietary guidance and public health strategies in Mexico [22]. Furthermore, other parameters, including various anthropometric measurements and their correlations with BMI, are utilized to assess the severity of obesity [23].

Obesity in Mexico has increased alarmingly. According to the 2020 ENSANUT, 75.2% of adults and 35.6% of children and adolescents are overweight or obese. This rise is attributed to various factors, including changes in eating habits, decreased physical activity, and socioeconomic factors.

In Mexico, obesity has risen dramatically, currently affecting more than 30% of the adult population [24]. Projections indicate that by 2050, obesity rates will exceed those of overweight individuals, with 54% of men and 37% of women being diagnosed with obesity [25,26]. This increase is driven primarily by increased consumption of high-calorie foods and a sedentary lifestyle. Furthermore, obesity is linked to several comorbidities, including cardiovascular disease, T2D, osteoarthritis, certain cancers, and obstructive sleep apnea, placing significant strain on Mexico’s healthcare system [27].

Excess weight is strongly linked to various disabilities across psychosocial, symptomatic, and work-related domains [28,29,30,31]. Studies indicate a correlation between higher BMI and increased pain, joint disorders such as osteoarthritis, fatigue, and sleep disturbances [32,33,34]. From a psychosocial standpoint, individuals with obesity are at greater risk for psychiatric conditions, particularly depression, and report a lower health-related quality of life (HRQoL) than those with a healthy weight [35,36]. Additionally, international research consistently shows that obesity is linked to decreased work productivity, with higher BMI levels associated with increased absenteeism and presenteeism.

Given the projected rise in obesity rates, understanding weight loss outcomes among overweight and obese individuals in Mexico is crucial. In Mexican adolescents, obesity treatment mainly focuses on improving diet and increasing physical activity through lifestyle changes, supported by various policy interventions implemented between 2018 and 2024 [37,38]. Weight loss strategies generally follow a stepwise approach, starting with dietary and lifestyle counseling and, when necessary, progressing to pharmacological or surgical interventions. Family involvement can enhance the effectiveness of weight loss efforts, particularly in family-oriented cultures.

Despite global initiatives aimed at combating obesity, achieving sustainable weight loss continues to be a challenge, as many individuals regain some or all of the weight lost within two years. A survey of Mexican university students revealed that nearly 40% were actively trying to lose weight [39]. However, large-scale data on the frequency of weight loss attempts among individuals with obesity in Mexico, their preferred methods, and their satisfaction with these methods remain limited [40].

### 2.2. Diabetes in Mexico

Diabetes can be defined as a condition with multiple causes that occurs when there is chronic hyperglycemia due to low or no production of the hormone insulin, as well as its ineffective use [41,42]. The American Diabetes Association (ADA) in 2023 has classified this condition to facilitate an accurate diagnosis of the disease (Table 2).

The development of diabetes in the human body primarily impacts the production and proper use of insulin (Figure 1). For example, type 1 diabetes (T1D) is characterized by the destruction of pancreatic β-cells, which are responsible for insulin secretion. This results in persistent hyperglycemia, stemming from various factors, such as viral infections, chemical agents, autoimmunity, or genetic predisposition [43]. In contrast, T2D involves a deficiency in insulin production due to dysfunction of β-cells. Furthermore, insulin resistance increases, leading to increased glucose production and/or decreased glucose uptake in insulin-dependent tissues.

The primary role of β cells is to secrete insulin, which helps decrease glucose production in the liver and promotes glucose uptake in adipose tissue and skeletal muscle. When insulin production decreases due to impaired β-cell function, hyperglycemia occurs, resulting from elevated blood glucose levels and decreased uptake [44]. According to the International Diabetes Federation (IDF), in 2021, an estimated 537 million adults, aged 20–79 years, were living with diabetes worldwide, with projections indicating that 643 million people will be affected by 2030. The global prevalence rate of diabetes in 2021 was 6.1%, making it one of the leading causes of death and disability [45]. Furthermore, the Pan American Health Organization (PAHO) reported that 284,049 people died from diabetes-related complications in 2009. That same year, diabetes became the second leading cause of years lived with disability and years of life lost due to premature death (Burden of Disease from Diabetes, n.d.).

The prevalence of diabetes in Mexico has consistently increased over recent decades, with T2D accounting for the majority of cases. In 2021, IDF ranked Mexico 7th among the top 10 countries with the highest rates of diagnosed diabetes, and it is projected to be 8th by 2045 [46]. The 2022 National Health and Nutrition Survey reported a diabetes incidence of 12.6%, with 11.3% among men and 13.6% among women. When categorized by age, the prevalence for the 20–39 age group was 2.2%, in contrast to the 10.6% prevalence reported by the ENSANUT (National Institute of Statistics) in 2020 for the same group, emphasizing an increase attributed to the impact of the COVID-19 pandemic [47].

Approximately 12 million Mexicans are living with diabetes, with a significant portion of the population remaining undiagnosed, worsening the issue [48]. These statistics indicate that women currently have a slightly higher prevalence than men do. Moreover, the risk of developing diabetes increases with age, peaking in individuals over 60 years of age, where the prevalence exceeds 20% [49,50]. The prevalence of diabetes also varies widely across different regions of Mexico, being higher in the central and northern areas, where urbanization and industrialization, along with related eating habits and lifestyles, are more prevalent [3,48]. As of April 2023, the Hospital Epidemiological Surveillance System for T2D reported a total of 10,102 admissions of patients diagnosed with T2D, with Tabasco (1561), Jalisco (998), and Puebla (720) having the highest number of cases.

Research has identified several risk factors associated with the development of diabetes in Mexico, highlighting the impact of environmental, social, and genetic influences [51,52]. Approximately 55% to 60% of adults in Mexico are overweight or obese, which is a major contributor to the onset of T2D [53]. Central obesity (abdominal fat) is particularly linked to insulin resistance, a precursor to diabetes [54,55]. The high consumption of processed foods and sugary beverages, coupled with a decreased intake of fruits, vegetables, and fiber, has increased the risk of diabetes in the country [56]. Economic access to ultra-processed foods and a lack of nutritional education further worsen this issue [57].

The transition to a more sedentary lifestyle, driven by urbanization and technological advancements, is another factor contributing to the increasing prevalence of diabetes [58]. Less than 50% of the adult population meets the recommended levels of physical activity [59]. Moreover, there is a racial predisposition, with indigenous and mestizo populations in Mexico exhibiting greater genetic susceptibility to T2D, which helps clarify the country’s high prevalence [60]. Poverty and low education levels also play significant roles, as they limit access to proper medical care, medications, and the adoption of healthy lifestyles [8]. Rural and indigenous communities encounter additional challenges in obtaining early diagnoses and receiving prompt treatment.

Diabetes is among the leading causes of mortality in Mexico, and its complications, including cardiovascular disease, kidney failure, neuropathies, and amputations, become chronic, placing a significant burden on the healthcare system. Studies have shown that complications related to diabetes lead to a marked reduction in quality of life and contribute to increased premature mortality [61,62].

In response to this crisis, the Mexican government has launched early detection programs and initiatives to encourage healthy lifestyles. Efforts such as the “Front of Label Warning” on food products and mass media prevention campaigns aim to reduce risk factors, although significant challenges remain in their implementation and effectiveness [63,64]. Moving forward, Mexico’s strategy for diabetes should prioritize prevention through stronger public health policies, improved nutrition education, and equitable access to medical services for vulnerable populations. Managing risk factors, along with the establishment of comprehensive disease management programs, is vital for mitigating the impact of diabetes in the country.

## 3. Relationships of the Intestinal Microbiota with Obesity and Diabetes

### 3.1. Intestinal Microbiota

The gut microbiota refers to the collection of microorganisms, primarily bacteria, that reside in the gastrointestinal tract of humans and other animals [65]. It has emerged as a crucial area of research because of its impact on human health and its connection to a wide array of diseases [66]. In recent years, studies on the gut microbiota have progressed substantially, investigating its composition, functions, and relationship with various pathological conditions [67].

The gut microbiota is composed primarily of bacteria from the Firmicutes and Bacteroidetes phyla, along with other groups such as Actinobacteria and Proteobacteria (Figure 2). The diversity of the microbiota can fluctuate depending on factors such as age, diet, environmental influences, and antibiotic use [68,69]. Diets high in fiber encourage the growth of short-chain fatty acid (SCFA)-producing bacteria, which are beneficial to health. Furthermore, the microbiota evolves and changes throughout life, exhibiting greater diversity in adulthood and diminished diversity in older individuals. Antibiotic use can also result in dysbiosis (an imbalance in microbial composition), which may lead to health issues, as this imbalance impacts critical functions of the microbiota that are essential for human health [67].

An example of this is the ability of the microbiota to help digest nondigestible compounds and produce essential vitamins such as B12 and K [70]. Additionally, the microbiota interacts with the immune system, facilitating the maturation of immune cells and the regulation of inflammatory responses [71]. Through the fermentation of dietary fiber by intestinal bacteria, the microbiota generates short-chain fatty acids, such as butyrate, which have anti-inflammatory and cancer-protective properties [72].

However, dysbiosis, or an imbalance in the composition of the intestinal microbiota, has been linked to various conditions, including Crohn’s disease and ulcerative colitis [73]. Recent studies also indicate that the microbiota plays a role in regulating energy metabolism and fat accumulation [74]. While not definitive, an increase in the proportion of Firmicutes compared with Bacteroidetes has been associated with obesity. Another related condition is diabetes, where the microbiota can impact both insulin resistance and glycemic control, contributing to the development of T2D [9]. Additionally, the “microbiota–gut–brain axis” has gained increasing attention, as it has been linked to disorders such as autism, depression, and Parkinson’s disease [75].

To restore the balance of the intestinal microbiota and achieve a state of eubiosis, probiotics and prebiotics have been utilized as intervention strategies [19]. Additionally, fecal microbiota transplantation (FMT) has been successfully applied to treat recurrent *Clostridioides difficile* infections and is currently being studied for other conditions, including obesity, metabolic syndrome, and ulcerative colitis [76]. However, despite these advancements, identifying the composition of the microbiota remains crucial. Sequencing and metagenomic analysis techniques have enabled a more precise and detailed characterization of the microbiota, resulting in the creation of databases for various bioinformatics studies [77,78]. Technologies such as next-generation sequencing (NGS) allow for the comprehensive identification of bacterial species and their functions [79]. Moreover, with the growth of personalized medicine, researchers are investigating how targeted manipulation of the microbiota could lead to more effective therapies tailored to individual patients. Current studies suggest that the gut microbiota may influence the pharmacokinetics of certain medications, such as immunosuppressants and anticancer drugs, affecting their efficacy and toxicity [80,81].

Despite significant progress, microbiota research faces several challenges, including the lack of standardized methodologies, which makes comparing study results difficult. Although many correlations between the microbiota and various diseases have been identified, clear causal relationships have yet to be established. As a result, while probiotics and prebiotics show promise, their effectiveness may be limited unless they are tailored to the unique characteristics of each patient’s microbiota. Thus, the intestinal microbiota represents an emerging field with critical implications for human health. Ongoing research is providing new insights into disease development and potential treatments. With the advent of advanced technologies and a personalized approach, microbiota research has the potential to open new frontiers in preventive and therapeutic medicine.

### 3.2. Diabetes and the Microbiota in Mexico

In Mexico, several studies have examined the connections among the gut microbiota, obesity, and diabetes. Research has highlighted significant differences in pediatric gut microbiota associated with T2D and metabolic syndrome, emphasizing their potential for developing microbiome-based predictive tools for children. Carrizales-Sánchez et al. [82] analyzed the gut microbiota of 66 children aged 7–17 years, who were divided into three groups—T2D (21), metabolic syndrome (25), and control (20)—via 16S rDNA gene sequencing. The findings revealed significant differences in microbiota composition between the groups at both the genus and bacterial family levels. Notably, changes in diversity indicated that distinct microbiota profiles were linked to each condition. *Faecalibacterium* and *Oscillospora* were more abundant in the metabolic syndrome group than in the control group, whereas *Prevotella* and *Dorea* tended to increase from the control group to the T2D group. Correlations with cardiometabolic factors revealed positive associations of *Prevotella*, *Dorea*, *Faecalibacterium*, and *Lactobacillus* with hypertension, abdominal obesity, high glucose, and elevated triglycerides. Consequently, specific microbial biomarkers were identified for each of the conditions studied.

Several studies have examined the relationship between the microbiota and diabetes in Mexican infants, indicating a complex interplay of genetics, diet, and early childhood exposures. Research shows that alterations in the microbiota may be connected to the development of T1D in high-risk populations. For example, studies involving Mexican children have identified dysbiosis (imbalanced microbial communities) as an early sign of T1D onset, particularly when it is influenced by factors such as perinatal conditions, diet, and antibiotic use [83].

In the case of T2D, studies have explored the potential role of the microbiota in obesity, a significant risk factor for the development of diabetes. In Mexican children, the composition of the microbiota, including the abundance of specific bacteria such as *Lactobacillus reuteri*, has been associated with an elevated risk of obesity, which significantly contributes to the onset of T2D. These findings underscore the importance of the gut microbiota in early childhood as a potentially modifiable factor for preventing or managing diabetes later in life [84].

Studies have focused on the pediatric population, which represents the most pressing issue in Mexico, being the country currently with the highest rate of this condition worldwide, and they have highlighted the problems exacerbated during the COVID-19 pandemic. The main causes of death in Mexico are linked to individuals with metabolic syndrome, particularly T2D [85]. Several research projects, including those conducted in Mexico, have focused on the connections among the gut microbiota, diabetes, and COVID-19, where the epidemiological context has provided valuable insights into how these factors impact public health. In patients with both COVID-19 and T2D, the gut microbiota shows alterations, particularly an increase in opportunistic pathogens, which may contribute to more severe disease outcomes [86].

In Mexico, the prevalence of both diabetes and COVID-19 has become alarming, especially among individuals with obesity and T2D, who are more vulnerable to the harmful effects of the virus. Previous studies indicate that dysbiosis in these populations may be a significant factor in the more severe progression of the disease [87]. In Mexican adults with T2D, dysbiosis has been identified in the distal colon microbiota, characterized by a relative increase in the abundance of *Bacteroidetes* compared with *Firmicutes* [88]. Similarly, high levels of specific bacterial taxa indicative of dysbiosis have been identified in diseases such as T2D in countries such as the United States and Brazil. These nations, along with Mexico, are among the three countries with the highest COVID-19 mortality rates [85].

The relationship between the gut microbiota and T2D in adults in Mexico has become an increasingly important area of research. The gut microbiota plays a crucial role in the development of metabolic diseases, such as T2D, by influencing processes such as nutrient absorption, modulating the immune system, and regulating energy balance. Specifically, patients with T2D exhibit an imbalance in their microbiota, known as dysbiosis, characterized by a decrease in beneficial bacteria such as *Bifidobacterium* and an increase in pathogenic species such as *Firmicutes* and *Proteobacteria* [89].

A 2021 study by Diener et al. [90] involving a cohort of 405 treatment-naive Mexicans at different stages of T2D identified bacterial genera as reliable biomarkers for disease incidence and risk. This research provides compelling evidence of the connection between the gut microbiome and T2D, demonstrating that alterations in the microbiome reflect the gradual progression of the disease. The authors reported that individual taxa were generally associated with specific clinical measurements. For example, the abundances of *Escherichia* and *Veillonella* increased with disease progression, with *Escherichia* being preferentially linked to blood glucose levels and *Veillonella* being associated with insulin-related measurements. Additionally, *Blautia* and *Anaerostipes* were observed to decrease with disease progression but were also linked to improved beta-cell function and insulin efficiency, marking the first time this relationship has been described.

A recent study by Lizárraga et al. [91] examined the changes in the fecal microbiota of Mexican women with gestational diabetes (GDM) and their newborns via 16S rRNA taxonomic fingerprinting. The research involved 17 women (8 with GDM and 9 controls) and 16 newborns, with fecal samples collected during the third trimester of pregnancy, at cesarean section, and from meconium samples from the newborns. The findings revealed that women with GDM had greater alpha diversity in their third trimester microbiota, showing a significant increase in taxa such as *Firmicutes*, *Lachnospiraceae*, and *Faecalibacterium*, which are linked to blood glucose levels. No significant differences were detected in the microbiota of the cesarean section samples between the groups. However, the newborns of mothers with GDM presented a significant increase in *Faecalibacterium* in their meconium microbiota, which was associated with the mothers’ BMI and fasting glucose levels. These results suggest that GDM is related to alterations in both the maternal fecal microbiota and the newborn meconium microbiota.

Similarly, Gámez-Valdez et al. [92] and Benítez-Guerrero et al. [93] examined the microbial composition of the microbiota in women with gestational diabetes, identifying specific species as potential biomarkers during pregnancy. Their research revealed that during the first trimester of pregnancy, *Proteobacteria* presented the highest relative abundance, followed by *Firmicutes* and *Actinobacteria*. In women with gestational diabetes, the abundances of *Achromobacter*, *Rhizobium*, *Bifidobacterium*, and *Mesorhizobium* were decreased. Conversely, greater abundances of UGC-014, *Clostridium_sensu_stricto_1* (class Clostridia), *Staphylococcus*, *Bosea*, *Rothia*, and *Enterobacter* were detected. Furthermore, in women with gestational diabetes, *Staphylococcus*, *Corynebacterium 1*, *Anaerococcus*, and *Prevotella* were found in greater proportions in the colostrum of those with obesity or GDM than in healthy women.

Although research on T2D and its relationship with the gut microbiota in the Mexican population is still limited, significant efforts have been made to identify correlations between intestinal dysbiosis and health conditions in this population. Studies investigating the gut microbiota in Mexico have highlighted the complexity and diversity of the intestinal bacterial communities and their strong links to various health issues, including dysbiosis. Identifying specific bacterial taxa and their roles in regulating gut health emphasizes the critical role of the microbiota in maintaining metabolic balance and preventing disease. Moreover, research has demonstrated that factors such as diet, lifestyle, and genetics significantly influence the gut microbiota, suggesting that interventions to restore microbial diversity may be essential for treating and preventing diseases related to dysbiosis in the Mexican population.

### 3.3. Obesity and the Microbiota in Mexico

Several studies have linked the gut microbiota to obesity, emphasizing its modulatory role in the gut–brain axis and its impact on appetite-regulating hormones such as leptin, ghrelin, and glucagon-like peptide 1. Another important factor is the neuronal connection between the gut and the brain via the vagus nerve, which is crucial for regulating eating behaviors and appetite [94,95].

Various mechanisms can influence appetite and contribute to the development of obesity. Specifically, appetite regulation is driven by the interaction between the central nervous system (CNS) and the endocrine system, where signals from peripheral organs, especially the digestive system, are transmitted to the CNS. Hormones such as leptin and ghrelin, which are produced in the periphery, play crucial roles in modulating eating behavior through their action on the hypothalamus [96]. Considering the above findings, the composition of the intestinal microbiota has been identified as a key environmental factor in regulating body weight and energy metabolism in individuals with obesity [97]. This association is linked to changes in the abundance of specific phyla, primarily Firmicutes and Bacteroidetes [98].

A study by Chávez-Carbajal et al. [99] involving 67 Mexican women revealed that Firmicutes was more abundant in women with obesity, a trend also observed in studies conducted with Austrian women. This phylum is thought to contribute to the development of obesity. Additionally, a greater presence of *Faecalibacterium* spp. from the *Ruminococcaceae* family was noted in the obese group. The study also reported an increased abundance of *F. prausnitzii*, which has been linked to obesity in certain populations. These findings suggest that factors such as ancestry, geographic location, and diet play crucial roles in microbiota composition and should be taken into account in future research.

In recent years, the role of the intestinal microbiota in obesity has become an increasingly important topic. Studies have shown that it influences energy extraction from the diet and eating behavior through various mechanisms, including regulating intestinal permeability, controlling inflammation, and releasing gut hormones [64]. Most research indicates that the diversity and richness of gut microbiota in individuals with obesity are reduced. As shown in Table 3, few studies have focused on the Mexican population, but they indicate a reduction in bacteria among patients with obesity compared with normal-weight subjects, highlighting different proportions of bacterial phyla, with an increase in Firmicutes and a decrease in Bacteroidetes.

In a previous study, Valsecchi reported that the microbiota of individuals with obesity was marked by an increase in the genera *Proteobacteria*, *Bacteroides*, *Campylobacter*, and *Shigella*, alongside a decrease in *Akkermansia muciniphila*, which is linked to anti-inflammatory effects. This dysbiosis contributes to the loss of mucosal barrier integrity, degradation of the intestinal mucosa, and increased oxidative stress [102].

As shown in the table, there was a decrease in the concentration of *Lactobacillus. Raoult* suggested that bacteria linked to obesity promote early lipid digestion, whereas beneficial species primarily digest simple sugars. A diet high in carbohydrates and fats undermines the intestinal barrier, allowing lipopolysaccharides from the membranes of Gram-negative bacteria to enter systemic circulation. This process triggers the release of proinflammatory cytokines, which are associated with obesity [64].

## 4. Interventions Aimed at Restoring the Microbiota

As highlighted, the gut microbiota plays a crucial role in the development and prevalence of various pathophysiologies [19]. Dysbiosis is a common factor in most conditions and, paradoxically, represents an opportunity to develop interventions aimed at restoring microbial balance [103]. One of the primary approaches has been dietary modification, which involves the incorporation of optimal nutritional components that are metabolized by gut microbiota microorganisms [104]. Similarly, probiotics and prebiotics have emerged as effective strategies for restoring the composition of the gut microbiota. Both interventions require careful consideration of multiple factors, including nutrient type, microbiota status, probiotic strain selection, and prebiotic source [105]. In Mexico, these interventions have been applied to evaluate various probiotic strains and specific dietary components that facilitate the transition from dysbiosis to eubiosis.

### 4.1. Diet Modification

Diet plays a crucial role in shaping the gut microbiota. Studies have shown that the intake of specific nutrients promotes microbiota diversification, encouraging the growth of beneficial bacteria such as *Bifidobacterium* and *Lactobacillus* [106]. In Mexico, dietary interventions aimed at improving microbiota composition in individuals with various conditions have yielded promising results, leading to increased metabolic marker levels and significant shifts in the gut microbiota composition [104,107]. The influence of food-derived metabolites on the gut microbiota occurs through several mechanisms (Figure 3), including antioxidant and hypocholesterolemic activities, the promotion of beneficial microorganisms, and the synthesis of short-chain fatty acids [108,109].

In conditions such as metabolic syndrome, which is characterized by a cluster of metabolic disturbances, a diet high in unsaturated fatty acids and functional foods has been shown to significantly improve various metabolic markers that mediate inflammation. These dietary changes affect systemic inflammation and result in positive shifts in the intestinal microbiota. Specifically, the composition of the microbiota is altered by lowering the *Prevotella/Bacteroides* ratio, a marker often linked to a proinflammatory state, and increasing the abundance of beneficial bacteria such as *Akkermansia muciniphila* and *Faecalibacterium prausnitzii* [107,108]. These species are crucial because they can produce SCFAs, particularly butyrate, which has strong anti-inflammatory effects.

SCFAs, such as butyrate, are crucial for regulating immune responses and maintaining intestinal integrity by promoting the health of epithelial cells and supporting gut barrier function. These SCFAs also play a role in modulating the expression of genes involved in inflammation and metabolism, further contributing to the positive effects of dietary interventions [107,108]. As these changes reflect improvements in both systemic inflammation and gut health, they underscore the potential of dietary strategies in managing metabolic syndrome and its associated complications. By modulating the gut microbiota and the subsequent production of SCFAs, such diets offer a promising therapeutic approach for addressing metabolic and inflammatory diseases.

Dietary fiber intake is crucial in shaping and diversifying the gut microbiota, which plays a significant role in overall health. In individuals with obesity, the composition of the gut microbiota is often imbalanced, marked by a decrease in beneficial bacteria and an increase in those that promote inflammation and metabolic dysfunction.

A study conducted on young Mexican adults with obesity revealed that increasing dietary fiber intake resulted in significant improvements in key anthropometric measures of obesity, including BMI and waist circumference. These positive changes were accompanied by alterations in the gut microbiota, notably increasing the abundance of beneficial bacteria, specifically *Lactobacillus* and *Bifidobacterium*. These genera are known for their role in maintaining gut health by fermenting fiber to produce SCFAs and promoting a favorable gut environment. Furthermore, *Lactobacillus* and *Bifidobacterium* have been linked to reduced intestinal inflammation and enhanced gut barrier function. By fostering the growth of these beneficial microbes, dietary fiber helps restore balance to the gut microbiota, supporting metabolic health and potentially alleviating obesity-related complications. These findings underscore the importance of dietary fiber in modifying the gut microbiota composition and improving metabolic parameters associated with obesity, suggesting its potential as a therapeutic strategy for managing obesity [110].

With respect to obesity, dietary interventions have increasingly concentrated on the intake of certain fiber-rich foods, such as prickly pear (*Opuntia ficus-indica*), which is widely found in Mexico and other Latin American countries. This vegetable is often eaten in various forms, including fresh, frozen, or precooked cladodes, which are frequently incorporated into salads or other dishes to increase their nutritional value [104,111]. Prickly pear has long been acknowledged for its beneficial properties, such as hypoglycemic, anti-inflammatory, antioxidant, and antihypercholesterolemic effects [112].

In murine models of obesity, cactus consumption has improved the gut microbiota composition via mechanisms that include protection against endotoxemia and the production of the protein occludin-1 in the colon, which is linked to reduced intestinal permeability [113]. This dietary intervention has been specifically tested in women diagnosed with obesity within the Mexican population, showing beneficial changes in the gut microbiota compared with those in women of normal weight. After consuming 300 g of cactus for 30 days, the results indicated that cactus fiber can stimulate the development of various microbial groups in individuals with obesity. Notably, there was an increase in bacteria from the Lachnospiraceae family, which are known for their fiber-degrading abilities [104]. Additionally, a reduction in Bacteroides was observed, likely due to the high concentrations of fructans in cactus, alongside an increased presence of bacteria from the genera *Roseburia* and *Eubacterium*, which are strongly associated with fiber intake [114].

In addition, nopal, including foods from the pre-Hispanic Mexican diet, has been shown to enhance metabolic health. Research indicates that combining nopal with other traditional foods, such as corn, black beans, squash, chia seeds, chili, and tomatoes, significantly reduces metabolic and cognitive abnormalities, as well as intestinal microbiota dysbiosis resulting from excessive sucrose intake in murine models [115]. Many of these foods contain prebiotic components that foster the growth of beneficial gut bacteria.

Compounds such as resistant starch type 2, arabinoxylans, xylooligosaccharides, monosaccharides, and oligosaccharides are particularly effective at stimulating the growth of beneficial microbes, including those from the *Lactobacillus*, *Bifidobacterium*, *Faecalibacterium*, and *Akkermansia muciniphila* genera. The increased presence of these beneficial bacteria, in turn, stimulates the production of SCFAs, including acetate, butyrate, and propionate, which help regulate the intestinal microbiota and maintain gut health [109]. These SCFAs are recognized for their anti-inflammatory properties and their role in maintaining intestinal barrier function, further highlighting the therapeutic potential of pre-Hispanic foods in managing metabolic and gastrointestinal health.

Among emerging prebiotics, the consumption of fiber from unconventional sources such as agave fiber has gained attention because of the diverse range of fructans that act as prebiotics and help reduce the permeability of the epithelial barrier [116]. The widespread distribution of various agave species in Mexico has identified these plants as excellent sources of fructans, minerals, phenolic compounds, and dietary fiber. Among the most studied prebiotics from the *Agave* genus are inulin-type fructans, which have been linked to hormonal modulation and other beneficial effects [117].

In addition to inulin, other bioactive compounds found in agave, such as agavins, have been linked to improvements in metabolic disorders. Agavins are particularly recognized for their ability to positively affect the intestinal microbiota, increase the production of short-chain fatty acids (SCFAs), and regulate the endocrine system [109]. Furthermore, saponins and sapogenins derived from agave plants are known to display a wide range of bioactivities, further increasing the potential of agave as a functional food for enhancing metabolic health [118]. These findings underscore agave’s multifaceted role in promoting gut health and metabolic well-being, making it a valuable dietary component.

Undoubtedly, diet plays a crucial role in regulating the gut microbiota. The consumption of foods that promote the growth of beneficial species and provide metabolites involved in regulating various metabolic diseases represents a promising avenue for improving health. However, the effectiveness of these foods needs thorough evaluation through clinical studies that assess their physiological activity and prebiotic effects. Additionally, it is essential to acknowledge the important role of food-derived prebiotics in supporting the survival and viability of probiotic microorganisms. Probiotics represent another strategy for modulating the gut microbiota, and their effectiveness can be enhanced when they are consumed alongside prebiotics, creating a synergistic approach to improve gut health.

### 4.2. Incorporation of Probiotics

The incorporation of probiotics into the diet has become one of the most widely researched strategies for modifying the intestinal microbiota. Probiotic consumption originally started with the intake of fermented foods, with yogurt being one of the most notable examples [119]. Elie Metchnikoff is regarded as a pioneer in examining the relationship between yogurt consumption and human health, emphasizing the beneficial bacteria found in this product. Currently, probiotics are acknowledged as safe alternatives for the treatment and prevention of various health conditions.

A probiotic is defined as a live microorganism that, when consumed in sufficient amounts, can stimulate the native microbiota, enhance innate immunity, and offer a variety of health benefits. However, recent research has shown that the metabolites produced by these microorganisms, known as post-biotics, also play crucial roles in promoting bioactivities that are beneficial to human health [120]. This growing body of knowledge indicates that the effects of probiotics extend beyond their viability. In fact, the concept of the “ghost probiotic” has emerged, referring to inactive or lysed probiotic cells that, despite lacking vitality, can still elicit beneficial effects in the host. This concept further expands the traditional understanding of probiotics and their potential advantages [121].

Probiotics are commonly integrated into the diet through the consumption of fermented foods and beverages, with fermented drinks being among the most widely consumed products globally [30]. In many cultures, consuming these products has been a part of the traditional diet, suggesting that they contain various probiotic strains and bioactive metabolites with potential health benefits [122].

The mechanisms of action exhibited by probiotic strains are varied (Figure 4) and play several significant roles in gut health. These roles include the production of bacteriocins and SCFAs, the regulation of the intestinal microbiota through colonization, and competition with pathogenic microorganisms for nutrient uptake and adhesion to the intestinal epithelium. Additionally, probiotics have immunomodulatory effects, regulating the concentration and expression of proinflammatory cytokines such as IL-10, NF-κB, and TNF-α. They also aid in reducing LPS endotoxemia and help maintain the integrity of the intestinal barrier by minimizing hyperpermeability [19,123].

In Mexico, more than 60 types of fermented beverages have been consumed regularly since the establishment of various civilizations [124]. Most of these beverages are made from substrates such as maguey sap, various grains, and fruits, resulting in drinks such as *pozol* and *pulque* [125]. Research has shown that consuming these beverages helps control the glycemic index and dyslipidemia, attributing these benefits to the probiotic strains present in their composition, including bacteria from the genera *Bifidobacterium* and *Lactobacillus* [125,126].

Research into probiotic interventions has been conducted within the Mexican population across various contexts, especially among patients with obesity and kidney disease. Under these conditions, the gut microbiota is typically characterized by a predominance of *Firmicutes* bacteria, reduced bacterial diversity, and the presence of taxa that affect multiple metabolic pathways [88,99]. Studies in Mexico have examined the effects of probiotics on the gut microbiota of overweight and obese children. One investigation revealed that consuming 15 mL of milk fermented with *Lactobacillus casei* (10^8^ CFU/mL) and enriched with *Agave salmiana* fructans for six weeks significantly increased microbial diversity and abundance in the gut microbiota [127]. Similarly, in patients with kidney disease undergoing hemodialysis, a probiotic intervention using a blend of *Lactobacillus acidophilus* and *Bifidobacterium bifidum*, along with 2.3 g of inulin taken three times daily for two months, resulted in an increase in *Bifidobacterium* spp. and significantly improved the maintenance and balance of their intestinal microbiota, highlighting the importance of symbiotic foods in therapeutic strategies [128].

In general, in vivo models of T2D, obesity, and hypertension have shown that probiotics can assist in insulin sensitization, regulate glucose transporters (GLUT4) and incretins (GLP-1), reduce LPS and proinflammatory cytokines, and maintain intestinal barrier integrity. These findings highlight the therapeutic potential of probiotics in managing metabolic and inflammatory conditions, offering promising evidence for their use as holistic treatment strategies.

Despite the promising results observed in vivo models, additional rigorous efforts are necessary to develop clinical studies that offer a comprehensive understanding of the mechanisms involved in regulating the intestinal microbiota through probiotic consumption and their impact on various pathophysiologies. Clinical studies on probiotics in Mexico remain limited, which hinders a thorough evaluation of the role these microorganisms play in metabolic disorders prevalent among the Mexican population. This gap presents a significant opportunity for research in the country aimed at evaluating different probiotic strains as potential adjuvants in the treatment and prevention of diseases such as obesity and T2D.

## 5. Challenges and Future Directions

Despite significant advances in research on the relationships between gut microbiota, obesity, and diabetes, several critical challenges remain that limit a comprehensive understanding and effective clinical application of these findings. One major obstacle is the individual variability in the gut microbiota composition, which is influenced by a range of genetic, environmental, and lifestyle factors. This variability means that what is effective for one individual may not necessarily work for another, complicating the development of standardized or universal treatments. Tackling this challenge requires a more personalized approach to probiotic interventions and microbiota-targeted therapies, taking into account the unique characteristics of each patient’s gut microbiota.

The gut microbiota, a complex community of microorganisms, varies widely among individuals due to numerous factors, such as diet, lifestyle, disease history, the use of antibiotics and other medications, and environmental influences, including the surrounding area. Each of these factors can significantly affect the structure and function of the gut microbiota, which in turn influences the body’s ability to regulate metabolism, inflammation, and other biological processes related to obesity and diabetes. Consequently, treatments that may be effective for a general population do not always produce the same results when applied to individuals with varying microbiological, genetic, and environmental characteristics. This variability underscores the need for more personalized approaches for treating metabolic diseases.

Furthermore, genetic factors play a critical role in how individuals respond to dietary and probiotic interventions. Genetics can impact the composition of the gut microbiota and how the body processes nutrients, as well as adaptations to changes in the gut environment. For example, certain genetic variants may predispose individuals to increased fat storage or insulin resistance, which can facilitate the development of obesity and T2D. As research into genetics and the gut microbiota continues to progress, more interactions between these two factors will be revealed, illuminating why some individuals are more prone to these diseases than others are. Understanding these genetic–microbiota interactions could ultimately lead to more personalized and effective treatment strategies for metabolic disorders. Many studies are based on small sample sizes, which reduces the statistical power and limits the generalizability of the results. In addition, most available data come from observational or cross-sectional designs, preventing the establishment of causal relationships between microbiota alterations and metabolic outcomes. Considerable methodological heterogeneity, such as differences in sequencing platforms, taxonomic classification pipelines, sample processing protocols, and analytical approaches, further complicates cross-study comparisons. The scarcity of longitudinal and well-controlled intervention studies also limits our understanding of temporal dynamics and the long-term effects of dietary, probiotic, or lifestyle interventions on the gut microbiota. These limitations underscore the need for more standardized, larger-scale, and mechanistically oriented studies to strengthen the evidence base moving forward.

Environmental influences, urbanization, diet, and exposure to pollutants significantly impact the microbiota. The modern diet, which is high in processed foods and low in fiber, plays a crucial role in shifting the gut microbiota, leading to an imbalance that encourages the growth of microorganisms associated with obesity and diabetes. Similarly, insufficient physical activity, chronic stress, and inadequate sleep are environmental factors that adversely affect the microbiota. Consequently, interventions that focus solely on one of these factors, without considering the complexity and interplay of all these elements, may prove ineffective.

Another major challenge in gut microbiota research and its effects on obesity and diabetes is the limited number of studies that can be effectively applied to diverse populations. The development of treatments that alter the microbiota, whether through probiotics, prebiotics, diet, or gene therapies, requires a deeper understanding of how different microbial compositions influence metabolic diseases. While broad approaches have shown some effectiveness, the variability between individuals and differences in geography, social factors, and cultural practices demand more tailored interventions. Addressing these factors will ensure that microbiota-based treatments are effective across various populations, emphasizing the need for personalized strategies in managing obesity and metabolic disorders.

These challenges are even more pronounced concerning the Mexican population. Mexico faces a high prevalence of obesity and T2D, conditions exacerbated by lifestyle changes, dietary habits, and various socioeconomic factors. The composition of the microbiota in Mexicans may significantly differ from that of other populations, implying that interventions successful elsewhere may not always be applicable or effective in this context. To address this, further studies on the gut microbiota specific to the Mexican population are essential, taking into account factors such as the traditional Mexican diet, access to healthcare, and local genetic predispositions. Gaining a deeper understanding of these factors will enable the development of more effective and personalized treatments tailored to the unique needs of the Mexican population. Current evidence on the relationships among the gut microbiota, obesity, and diabetes presents several methodological limitations that restrict the interpretability and comparability of findings.

Moreover, multidimensional approaches that integrate personalized diets, synbiotic supplementation (a blend of probiotics and prebiotics), and other complementary therapies seem to offer a promising route toward more effective interventions. However, these solutions must be supported by strong evidence from large-scale clinical trials to prove their safety, efficacy, and, most importantly, relevance to the daily lives of affected individuals. Only through thorough research and practical application can these strategies be customized to meet the diverse needs of individuals, ensuring their success in managing obesity, T2D, and related metabolic disorders.

With respect to future directions, research on the relationships among the gut microbiota, obesity, and diabetes must continue to address several areas:Genomic and microbiota research: More detailed studies on how human genes interact with the gut microbiota are essential. Advances in identifying genetic biomarkers that can predict susceptibility to obesity and diabetes are needed, as well as understanding how these biomarkers are related to the composition of the gut microbiota.Personalized interventions: The future of probiotic and prebiotic therapies is rooted in personalization. Advances in precision medicine will enable interventions to be tailored specifically to each individual’s microbiological and genetic traits. This research will also involve the development of data-driven treatment strategies, where gut microbiota studies are integrated with genetic and health data to customize the therapeutic approach.Dietary and synbiotic treatments: Diet and synbiotic supplements are anticipated to play crucial roles in managing obesity and diabetes, although optimal combinations for each individual need to be identified. Long-term intervention studies are necessary to evaluate the impact and sustained effectiveness of these treatments.Integrative approach: In addition to the microbiota, further research is necessary to understand how additional factors, including socioeconomic status, lifestyle, and patient psychology, influence the gut microbiota and, in turn, obesity and diabetes. An integrative approach that takes all these factors into account will be vital for developing more comprehensive and effective prevention and treatment strategies.Advancing the development of new animal and clinical models is essential. Studies involving animal and human models must progress to more accurately replicate the conditions of human patients. Innovative experimental models may provide more detailed data on how the microbiota interacts with metabolism and diseases associated with metabolic syndrome.

In conclusion, although there has been significant progress in understanding the relationships among the microbiota, obesity, and diabetes, the journey to developing effective, personalized treatments remains lengthy. Future research should aim to address challenges associated with individual variability, identify personalized interventions, and explore factors specific to the Mexican population to offer more precise and effective solutions for treating these widespread conditions.

## Figures and Tables

**Figure 1 nutrients-17-03661-f001:**
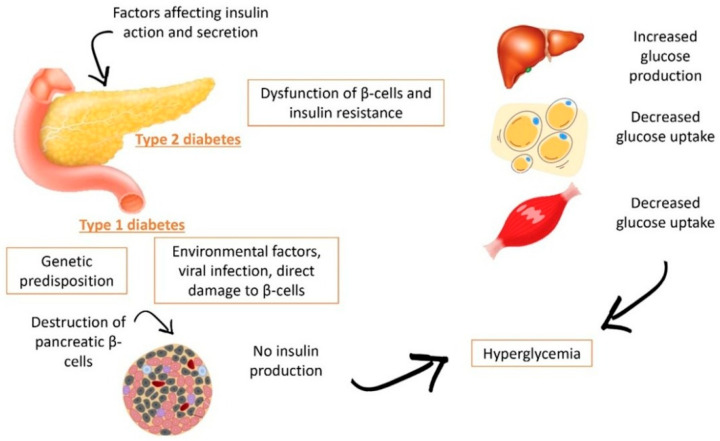
Schematic comparison of the pathogenic processes underlying type 1 diabetes (T1D) and type 2 diabetes (T2D). T1D is driven by autoimmune destruction of pancreatic β-cells, often triggered by genetic and environmental factors, leading to an absolute lack of insulin and impaired glucose uptake from early stages. In contrast, T2D combines β-cell dysfunction with insulin resistance in key tissues, such as liver, muscle, and adipose tissue, resulting in increased endogenous glucose production and reduced peripheral glucose utilization. Under both conditions, these alterations converge in persistent hyperglycemia.

**Figure 2 nutrients-17-03661-f002:**
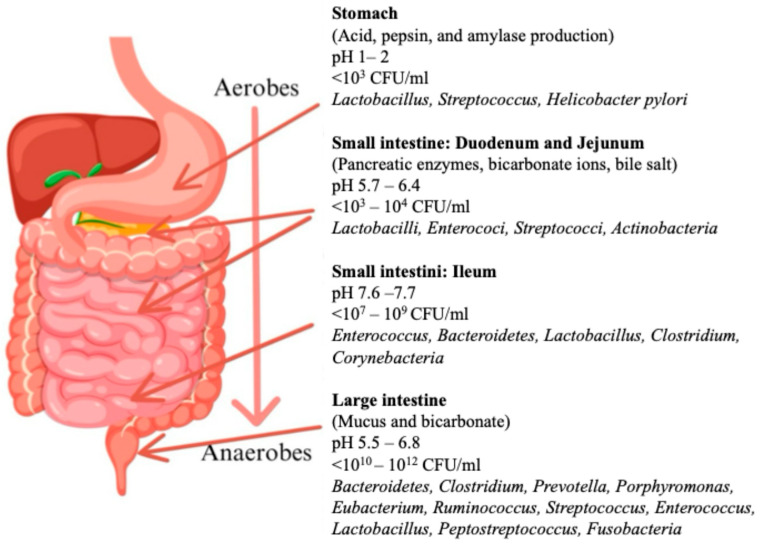
Distribution and general characteristics of the microbiota along the human gastrointestinal tract. Overview of the main gastrointestinal regions of the stomach, small intestine (duodenum, jejunum, and ileum), and large intestine showing their typical pH, microbial load, and predominant bacterial genera. The stomach presents highly acidic conditions and low microbial density, dominated by *Lactobacillus*, *Streptococcus*, and *Helicobacter pylori*. Microbial abundance increases throughout the small intestine, with moderate loads and genera such as *Lactobacillus*, *Enterococcus*, *Streptococcus*, and Actinobacteria. The ileum has a relatively high density and slightly alkaline pH, with notable presence of *Enterococcus*, Bacteroidetes, *Clostridium*, and *Corynebacterium*. The large intestine has the greatest microbial richness and density and is dominated by strictly anaerobic genera, including Bacteroidetes, *Clostridium*, *Prevotella*, *Ruminococcus*, and *Eubacterium*, along with some facultative anaerobes, such as *Enterococcus* and *Streptococcus*.

**Figure 3 nutrients-17-03661-f003:**
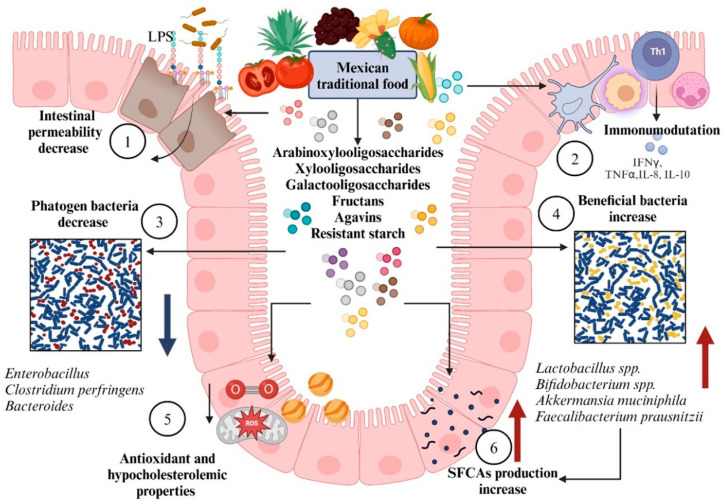
Mechanisms of intestinal microbiota regulation through metabolites derived from various foods. (1) Decreased intestinal permeability and strengthening of the epithelial barrier, limiting the translocation of lipopolysaccharides (LPS) and preventing metabolic endotoxemia; (2) immunomodulatory activity, promoting a reduction in proinflammatory interleukins and supporting a more balanced immune response; (3) a reduction in the concentration of pathogenic bacteria through direct antimicrobial effects or by creating conditions unfavorable for their growth; (4) an increase in beneficial bacterial communities that compete for adhesion sites on the epithelium, thereby limiting colonization by pathogenic species; (5) antioxidant and hypocholesterolemic properties that help reduce oxidative stress and the atherogenic index; and (6) increased production of short-chain fatty acids such as acetate, propionate, and butyrate, which are generated by beneficial bacteria and associated with the maintenance of intestinal homeostasis.

**Figure 4 nutrients-17-03661-f004:**
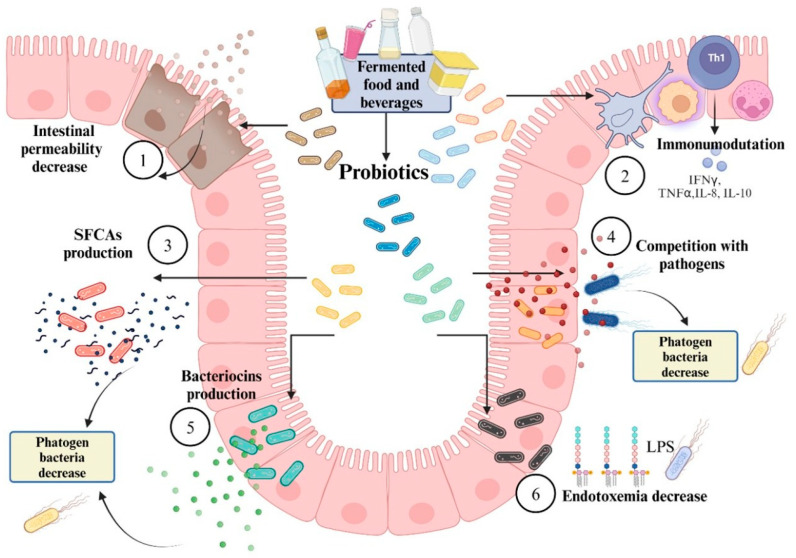
Mechanisms of intestinal microbiota regulation through probiotic consumption. (1) Decreased intestinal permeability and reinforcement of the epithelial barrier support the integrity of mucosal defenses; (2) immunomodulatory activity contributes to the reduction of proinflammatory interleukins and promotes a more controlled immune response; (3) production of short-chain fatty acids (SCFAs), which play key roles in energy metabolism and intestinal homeostasis; (4) competition with pathogenic microorganisms for adhesion sites and nutrient sources, limiting their colonization; (5) production of antimicrobial substances that inhibit or reduce the growth of pathogenic bacteria; and (6) increased endotoxemia through the inhibition of bacterial toxins, including lipopolysaccharides (LPSs), thereby helping maintain metabolic balance.

**Table 1 nutrients-17-03661-t001:** Obesity classification according to NOM-043-SSA2-201.

Underweight	Normal	Overweight	Obesity		
			Grade I	Grade II	Grade III
<18.5	18.5–24.9	25.0–29.9	30.0–34.9	35.0–39.9	>40.0
Abdominal Obesity According to the Official Mexican Standard NOM-043-SSA2-2012					
Male			Female		
<90 cm			<80 cm		

**Table 2 nutrients-17-03661-t002:** Classification of diabetes.

Type	Definition	Signs and Symptoms
Type 1 Diabetes	There is an autoimmune destruction of the pancreatic β-cells, resulting in an absolute insulin deficiency	Polyuria, Polydipsia, Polyphagia, Excessive fatigue, Blurry vision, Cuts/bruises that do not heal easily, Weight loss (type 1), Tingling, pain, or numbness in the hands and feet (type 2) (American Diabetes Association)
Type 2 Diabetes	Caused by a progressive deficit in insulin secretion from the pancreatic β-cells initiated by an insulin resistance state	
Gestational Diabetes	Diagnosed in the second or third month of pregnancy and was not present before gestation	Increased thirst and the sensation of urinating
Specific Types of Diabetes	Such as monogenic diabetes syndromes, exocrine pancreatic diseases, and drug or chemical-induced diabetes	Not applied

**Table 3 nutrients-17-03661-t003:** Relationship between microbiota and obesity.

Study	Population	Conditions Investigated	Microbiota Characteristics in Obesity	Other Relevant Findings	Reference
**Cross-sectional analytical study**	67 Mexican women aged 18 to 59 years, without antibiotic treatment	Obesity and obesity plus Mets	The Firmicutes phylum was more abundant in women with obesity. Additionally, a higher number of *Faecalibacterium* spp. taxa from the *Ruminococcaceae* family were observed in this group.	Predicted functional pathways associated with MetS: altered carbohydrate metabolism, endotoxemia, and inflammatory pathways	[99]
**Comparative cross-sectional study**	65 male and female Mexican volunteers aged 18–59 years with BMI > 29.9 kg/m^2^	Obesity and intestinal dysbiosis	The Proteobacteria/Firmicutes ratio significantly increased in the obesity group, with a predominance of aerobic and Gram-negative bacteria. The class Negativicutes and the genus *Lachnoclostridium* were associated with the obesity group, along with the *Streptococcaceae* family (order Lactobacillales) and *Enterobacteriaceae*.	Associations between dysbiosis, pro-inflammatory profile, folate (B9) disturbances, and altered carbohydrate metabolism.	[100]
**Comparative analytical study**	64 young Mexican volunteers, without prior medical treatment with antibiotics, probiotic or prebiotic supplementation, or chronic diseases	Obesity and metabolic inflammation	Young individuals with obesity had a lower total bacterial count compared to the normal BMI group. Specifically, those with obesity showed higher amounts of *Clostridium leptum* and *Lactobacillus*, and lower amounts of *Prevotella* and *Escherichia coli*.	Endotoxemia correlated with IL-6 and other inflammatory markers. Early obesity linked to systemic low-grade inflammation.	[101]
**Case–control studies**	66 subjects aged 7 to 17 years with T2D and metabolic syndrome, divided into three groups: (a) T2D (21), (b) metabolic syndrome (25), and (c) controls (20)	Metabolic syndrome and type 2 diabetes in children	The T2D group showed a peculiar presence of Succinibrionaceae. Another abundant genus in both T2D and metabolic syndrome was *Prevotella*. Specifically, for the T2D group, two genera were found in lower abundances: *Lactobacillus* and *Succinivibrio*.	Metabolic alterations associated with early inflammatory profiles.	[82]

BMI: Body mass index; T2D: Type 2 diabetes.

## Data Availability

No new data were created or analyzed in this study.

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
