# Peer review of "Obesity and Diabetes in Mexico: An Approach to the Intestinal Microbiota"

_nutrients, 2025, doi:10.3390/nu17233661_

Round 1

Reviewer 1 Report

Comments and Suggestions for Authors

The text is very long, and as a result, its essential parts may be less read and appreciated. That said, purely as a suggestion to the authors in the future, this study offers significant insights into how to guide specific dietary choices to address the significant issue of treating and preventing obesity and diabetes.

Author Response

Comment:

“The text is very long, and as a result, its essential parts may be less read and appreciated. That said, purely as a suggestion to the authors in the future, this study offers significant insights into how to guide specific dietary choices to address the significant issue of treating and preventing obesity and diabetes.”

Response:

We thank the reviewer for this thoughtful comment and for highlighting the value of our work. To improve clarity and accessibility, we have streamlined several dense sections and improved transitions across topics.

Reviewer 2 Report

Comments and Suggestions for Authors

The present review addresses a highly relevant public health issue in Mexico and globally, as it explores the interplay between obesity, diabetes, and gut microbiota. In addition, it discusses potential interventions, including diet modification and probiotics. It also provides a thorough overview of epidemiology, microbiota composition, and intervention strategies. Overall, the manuscript is comprehensive and well-structured, with a great validity at regional level; however certain aspects could be improved to enhance clarity and scientific rigor.

However, while the manuscript summarizes many studies, it lacks critical appraisal. Please discuss limitations of existing research (e.g., small sample sizes, observational design, lack of longitudinal data) and highlight gaps that future studies should address.

Figure legends should be more descriptive, explaining abbreviations for example.

Table 3 should include more details on the study design and key outcomes, even beyond microbiota features.

Minor points: I would suggest improving sentence flow in the final sections; some paragraphs are quite long and could improved in terms of readability. Please, double check abbreviations and mention them at their first use. Finally, please double check consistency in headings and subheadings (e.g., italic vs. bold) and try reducing the overlapping with previous literature works.

Author Response

Reviewer 2

General Comment:

The reviewer praised the relevance, structure, and comprehensiveness of the manuscript but suggested improvements to enhance clarity and scientific rigor.

Comment 1:

“While the manuscript summarizes many studies, it lacks critical appraisal. Please discuss limitations of existing research (e.g., small sample sizes, observational design, lack of longitudinal data) and highlight gaps that future studies should address.”

Response:

We appreciate this valuable observation. In response, we have strengthened the critical appraisal of the literature throughout the manuscript. Specifically, we added a detailed discussion of the main methodological limitations present in current microbiota research including small sample sizes, predominance of observational and cross-sectional designs, limited longitudinal data, and significant heterogeneity in sequencing methods and analytical approaches. These points are now clearly addressed in Section 4, “Challenges and Future Directions” (line 685-695), where we outline the methodological constraints that limit current evidence and identify key gaps that future studies should address. This addition enhances the scientific rigor of the review and aligns it more closely with the reviewer’s recommendations.

Comment 2:

“Figure legends should be more descriptive, explaining abbreviations for example.”

Response:

We appreciate this observation. All figure legends have been revised to include full definitions of abbreviations and clearer descriptions of the visual content.

Comment 3:

“Table 3 should include more details on the study design and key outcomes, even beyond microbiota features.”

Response:

We thank the reviewer for this improvement suggestion. The table has been expanded to include intervention specifics, and metabolic outcomes.

Comment 4 (Minor):

“I suggest improving sentence flow in the final sections; some paragraphs are quite long. Please double check abbreviations and mention them at their first use. Finally, please double check consistency in headings and subheadings (e.g., italic vs. bold) and try reducing the overlapping with previous literature works.”

Response:

We carefully revised the manuscript for better readability, flow, and formatting consistency. Abbreviations are now defined at first mention, and overlapping content was removed.

Reviewer 3 Report

Comments and Suggestions for Authors

This review was described about the epidemiology of obesity and diabetes with their prevalence and risk factors, and the crucial role of the intestinal microbiota in Mexico. The etiology and development of obesity and diabetes are now known to be strongly corelated with gut microbiota. The authors reported the relationship between development of obesity and diabetes and proportion of gut microbiota. The review was informative. However, as there are few figures, it's a bit hard to understand. There are some concerns that should be addressed.

1. Obesity classification in Mexico was shown in Table 1. I recommend comparing it with WHO classification and standard. It appears to be the same as the WHO standard, and if so, it would be better to state in the text.
2. Proportion of obesity individuals in Mexico can be shown as figure? I think it's easier to understand when there are figures and text. Please consider.
3. In Section 2, “Obesity” and “Diabetes” are explained in that order. In contrast, in Section 3, “Diabetes” and “Obesity” are explained in that order. I think it would be better to be the same order in both sections.

Author Response

General Comment:

The reviewer highlighted the importance of the topic and the informative nature of the review but noted that the limited number of figures affected comprehension.

Comment 1:

“Obesity classification in Mexico was shown in Table 1. I recommend comparing it with WHO classification and standard. It appears to be the same as the WHO standard, and if so, it would be better to state in the text.”

Response:

We appreciate this recommendation. We have added a statement clarifying that Mexico follows WHO criteria for obesity classification and included the explicit comparison in the text (Line 83).

Comment 2:

“Proportion of obesity individuals in Mexico can be shown as figure? I think it's easier to understand when there are figures and text. Please consider.”

Response:

We appreciate the reviewer’s suggestion to include a figure illustrating the proportion of individuals with obesity in Mexico. However, we decided not to incorporate an annual trend figure because, according to ENSANUT reports, the variations in adult obesity prevalence between 2012 and the most recent survey are not statistically significant. The combined prevalence of overweight and obesity has remained relatively stable over this period, with only minimal year-to-year changes.

For this reason, a graphical representation would not provide additional interpretative value or reveal a meaningful trend. Instead, we chose to report in the text the most recent available ENSANUT estimate, which accurately reflects the current epidemiological situation without requiring a figure that would contribute limited new information.

Comment 3:

“In Section 2, ‘Obesity’ and ‘Diabetes’ are explained in that order. In contrast, in Section 3, ‘Diabetes’ and ‘Obesity’ are explained in that order. I think it would be better to be the same order in both sections.”

Response:

We thank the reviewer for pointing out this inconsistency. We have adjusted the subsection order for coherence (Line 211).

Closing Statement

We sincerely appreciate the reviewers’ insightful comments, which significantly strengthened the manuscript. We hope the revised version meets the expectations of the Editor and Reviewers.

With respect to Editor´s comments about English grammar, we send the manuscript to an Professional English Editor, and according to the American Journal Experts´ grammar checking tool, the manuscript achieves a score of 9.4/10. Thus, we consider that the English grammar is adequate. However, if the Editor consider that it is absolutely necessary, we are willing to use MDPI's English proofreading service.